# Enhanced Physical and Mechanical Properties of Nitrile-Butadiene Rubber Composites with *N*-Cetylpyridinium Bromide-Carbon Black

**DOI:** 10.3390/molecules26040805

**Published:** 2021-02-04

**Authors:** Egor A. Kapitonov, Natalia N. Petrova, Vasilii V. Mukhin, Leonid A. Nikiforov, Vladimir D. Gogolev, Ee Le Shim, Aitalina A. Okhlopkova, Jin-Ho Cho

**Affiliations:** 1Department of Chemistry, North-Eastern Federal University, 677000 Yakutsk, Russia; kapitonov281087@mail.ru (E.A.K.); mvvnj@yandex.ru (V.V.M.); leonik87@mail.ru (L.A.N.); lightykt@mail.ru (V.D.G.); okhlopkova@yandex.ru (A.A.O.); 2School of Mechanical and Automotive Engineering, Wonju 26404, Korea; elshim@halla.ac.kr

**Keywords:** nitrile-butadiene rubber composite, carbon black, tensile strength, volumetric wear, planetary mill

## Abstract

The physical and mechanical properties of nitrile–butadiene rubber (NBR) composites with *N*-cetylpyridinium bromide-carbon black (CPB-CB) were investigated. Addition of 5 parts per hundred rubber (phr) of CPB-CB into NBR improved the tensile strength by 124%, vulcanization rate by 41%, shore hardness by 15%, and decreased the volumetric wear by 7% compared to those of the base rubber-CB composite.

## 1. Introduction 

Nitrile-butadiene rubber (NBR) exhibits a relatively low density, moderate tensile strength, and high oil resistance [1,2]. A higher acrylonitrile content in NBR leads to higher heat resistance, gas impermeability, and tensile strength; however, it also decreases elasticity and frost resistance [3]. Although a lower acrylonitrile content in NBR generally induces frost resistance in the rubber composite, it is insufficient to ensure reliable operation of various machinery and equipment under extremely low temperatures in locations such as Siberia and the Arctic Circle. Therefore, it is vital to ensure adequate levels of frost resistance and other physical and mechanical properties of NBR composites in extremely cold environments.

Carbon black has been utilized in NBR composites as a common reinforcing filler, particularly in sealing parts [4]. Consequently, various physical or chemical modifications of carbon black surface have been investigated to improve the physical and mechanical characteristics of rubber materials. Wu et al. employed a high-energy electron beam irradiation technique on carbon black at different doses to enhance the mechanical properties of natural rubber composites [5]. Park and Kim chemically modified carbon black surfaces to increase the tearing energy of butadiene rubber composites [6]. Plasma surface modification of carbon black in an SBR matrix has also been reported [7]. However, these physical or chemical modifications are not practical for industrial scaling because of their high cost and low efficiency. 

Mechanical activation is an economical method for carbon black activation [8]; it increases the specific surface area of carbon black particles and enhances the interaction between the filler and NBR, which eventually leads to improved physical and mechanical properties in the NBR composites [8]. Addition of surfactants is another economical carbon black activation method. A surfactant improves the uniform dispersion of carbon black and other additives during compounding and can enhance the vulcanization rate to significantly improve the physical and mechanical properties of the rubber composites [9,10,11]. For instance, the cationic surfactant cetylpyridinium chloride improves the processability and cure rate of epoxy group-containing acrylic elastomers [12]. Joint mechanical activation [13] of carbon black with a surfactant can maximize the uniform distribution of components in the matrix, which can eventually lead to enhanced physical and mechanical properties of the composites.

In this study, a practical and effective carbon-black modification technique, involving its joint mechanical activation with a surfactant using a planetary mill, was evaluated to improve the physical and mechanical properties of NBR composites for applications in extremely cold environments.

## 2. Results and Discussion

Vulcanization, physical, mechanical, and swelling properties of the BNKS-18 rubber composites were determined using the techniques discussed in the next section, and are summarized in Table 1. The cure rate indices (CRI) of composites **4** and **5** increased by 140% and 9%, respectively, compared to that of composite **1**. The difference in vulcanization rates of rubber composites **4** and **5** can be explained by the significant increase in the surface area of composite **4** owing to the addition of CPB-CB.

The addition of CPB–CB also increased the tensile strength (*f*_p_) and decreased the elongation at break (ε_p_) value of the BNKS-18 rubber composites, as shown in Table 1. The tensile strength and elongation at break of composite **2** (containing 5 phr of CPB-CB) increased by 124% and decreased by 34%, respectively, compared to those of composite **1**.

The tensile stress at 100% elongation (*f_100_*) and tensile strength (*f*_p_) of rubber composites **2**–**4** (with CPB-CB) increased significantly compared to those of composite **1** (without CPB-CB). The shore hardness of the rubber composites **2**–**4** increased by ~15% compared to that of composite **1**, regardless of the addition of CPB-CB. The degree of swelling for all the rubber composites (**2**–**4**) decreased slightly compared to that of composite **1**. The rubber composite **5** (with carbon black and base rubber) exhibited similar vulcanization, physical, and mechanical properties as those of the rubber composite **1**. This implies that the simple activation of carbon black does not cause any significant change in the investigated properties of the rubber composites.

The enhanced tensile strength (*f*_p_) and tensile stress at 100% elongation (*f_100_*) of rubber composites **2**–**4** are owing to the added CPB-CB. The surfactant CPB behaves as a dispersant and promotes the formation of new phase boundaries during the activation of carbon black. The investigation of a carbon black and surfactant model has shown that the surfactant actively interacts with the carbon black particles in aqueous solutions [14]. According to this model, the enhanced hydrophilicity of the CB particle surfaces leads to higher stability in the aqueous suspension and smaller CB particles. Because similar processes can occur in the solid phase of the rubber mixtures, the introduction of a preferably polar organic surfactant (CPB) to CB induces enhanced dispersion of the aggregated CB and hydrophilization of the CB surface in rubber composites [15]. Therefore, a better interaction between the surfactant-modified-carbon black (CPB-CB) and BNKS-18 rubber results in a more uniform filler distribution.

The adsorptive decrease in material strength (Rehbinder effect) might occur during the joint activation of CB with the surfactant CPB in a planetary mill [16,17]; therefore, the surfactant is more effective in reducing any possible aggregation compared to mechanical activation without a surfactant. Therefore, the surfactant helps to achieve a higher degree of dispersion of CB in rubber compounds [9,18,19].

The abrasion characteristics obtained using the universal friction machine at a speed of 30 cm/s are shown in Figure 1.

Figure 1a shows the slightly decreasing volumetric wear of the rubber composites with added CPB-CB of up to 10 phr (composites **2**–**3**), followed by an increase at 15 phr (composite **4**). The optimal properties were obtained from the rubber composite **2** containing 5 phr of CPB-CB. A further increase in the CPB-CB content causes a gradual decrease in tensile strength and elongation at break as summarized in Table 1. Figure 1b shows the maximum friction coefficient of rubber composite **2** corresponding to the lowest volumetric wear (in Figure 1a) and the highest tensile strength (Table 1). The improved wear resistance can be explained by a general increase in the material tensile strength, its modulus, and an even distribution of all compounds in the composite.

Figure 2 shows the SEM images of the surfaces of the rubber composites after abrasion. The BNKS-18 rubber composites exhibit distinct and consecutive transverse ridges along the sliding direction. The ridges are equidistant, indicating a high friction between the low modulus rubber and the counter surface and the influence of fatigue and abrasive wear, which are the two elements of rubber friction [20,21]. Detached crests are visible as debris on the rubber surface.

The internal structures of ruptured surfaces of the rubber composites were also investigated by SEM, and the resulting images are shown in Figure 3.

Figure 3 (2a,2b) show a rubber composite sample with 5 phr CPB-CB characterized by a less distinctive microrelief. This can be explained by the better distribution of compounds owing to CPB-CB addition. For the rubbers modified by CPB-CB (Figure 3 (2a–4b)), the electron micrographs show more uniform phase morphologies and a more even distribution of compounds over the material, which ensure enhanced physical and mechanical properties of the composites.

## 3. Materials and Methods

A BNKS-18 grade NBR (AO KZSK, Krasnoyarsk, Russia; specification: TU 38.30313-2006), which is a copolymer of butadiene and 18% acrylonitrile, was employed as a base rubber owing to its adequate hydrocarbon resistance and for having the lowest glass transition temperature among NBRs (T_G_ = −46 °C). A general purpose N550 carbon black (OAO Ivanovskii Tekhuglerod I Rezina, Ivanovo, Russia; standard: ASTM D1765-18) was utilized owing to its common use in ordinary rubber products, including tires. N-cetylpyridinium bromide (CPB, ZAO Vekton, St. Petersburg, Russia; specification: TU 6-09-09-70-77) was employed as a cationic surfactant.

Joint mechanical activation of CPB and carbon black was carried out in an Activator-2S planetary ball mill (OOO Mashinostroitelnyi zavod “Aktivator”, Novosibirsk, Russia) for high-efficient activation with a high dispersion degree and possible mechanochemical reactions [22,23,24]. The activation parameters were: rotational speed of the central axis: 700 rpm, drum rotation speed: 1050 rpm, centrifugal acceleration up to 550 m/s^2^, grinding ball weight: 300 g, amount of carbon black loaded into each drum: 30 g, and amount of CPB loaded into each drum: 3 g (9.1 wt%). The activation duration was 90 s. The resulting product was designated as CPB–carbon black (CPB-CB). Carbon black was also activated under the same conditions without CPB for comparison.

The composition of the base rubber for comparison was the following: 100 phr (parts per hundred rubber) of BNKS-18 rubber, 2.5 phr of zinc oxide, 2.5 phr of sulfur, 1.5 phr of 2-mercapto-benzothiazole (MBT), and 1.5 phr of stearic acid [25]. The base rubber was combined with various amounts of carbon black in a Plastograph EC Plus mixer (Brabender GmbH & Co., KG, Duisburg, Germany) at 40 °C and 40 rpm with a Banbury rotor for 20 min. NBR composites with CPB-CB were generated by combining the carbon black containing the base rubber with CPB-CB in a Polymix 110 L laboratory mill (Brabender GmbH & Co., KG) at 25 °C and 5 rpm for 10 min. As summarized in Table 2, the addition of CPB-CB was implemented by replacing the initial carbon black amount up to 15 phr and keeping the total amount of fillers at 50 phr in all cases. After 24 h, the rubber composites were vulcanized at 155 °C in a hydraulic press (GT-7014-H10C, GOTECH, Taichung, Taiwan) for 20 min.

Vulcanization characteristics were measured at 155 °C in an RPA-2000 rheometer (Alpha Technologies, Hudson, NY, USA). Various physical and mechanical properties, such as tensile strength and elongation at break, were determined according to the Russian standard GOST 270-84 [26] with a universal testing machine (Autograph AGS-J, Shimadzu, Kyoto, Japan) at 25 °C and a strain rate of 500 mm/min. The swelling degree was evaluated using GOST 9.030-74 [27] in Hydraunycoil FH51 oil (specification: MIL-H-5606F) at 70 °C for 72 h. The swelling degree was determined by the following equation:*Q* = [(*m_f_* − *m_i_*)/*m_i_*] × 100%(1)

*m_f_*: final mass of sample after swelling; *m_i_*: initial mass of sample.

Wear resistance during abrasive wear was determined using a universal friction machine (UMT-3, CETR, Bruker, Billerica, MA, USA) with a “pin-on-disk” friction scheme with a sliding speed of 30 cm/s, a P180 abrasive at 5 N load, and cylindrical samples with a height and width of 10 mm. Shore hardness was also measured with the universal friction machine. The structural characteristics of the rubbers were investigated by scanning electron microscopy (SEM) using JSM-6480LV (JEOL, Tokyo, Japan).

## 4. Conclusions

The effects of *N*-cetylpyridinium bromide (CPB) on carbon black (CB) modification were investigated in a planetary mill. The addition of 5 phr of CPB-CB into nitrile-butadiene rubber (NBR) improved the tensile strength by 124%, vulcanization rate by 41%, and shore hardness by 15%, and decreased the volumetric wear by 7% compared to those of the base rubber-CB composite; however, the degree of swelling did not change appreciably. This improvement resulted from enhancement in the even distribution of all compounds in the composite owing to CPB-CB. The modification of CB with CPB is more practical, efficient, and economical compared to other chemical or physical modifications of CB, and can therefore be applied in frost-resistant rubber sealing parts in heavy machinery operating in extremely cold environments, such as the Arctic regions.

## Figures and Tables

**Figure 1 molecules-26-00805-f001:**
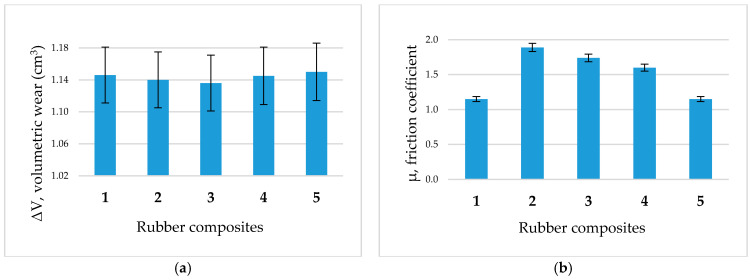
Abrasion characteristics of the rubber composites. (**a**): volumetric wear, (**b**): friction coefficient.

**Figure 2 molecules-26-00805-f002:**
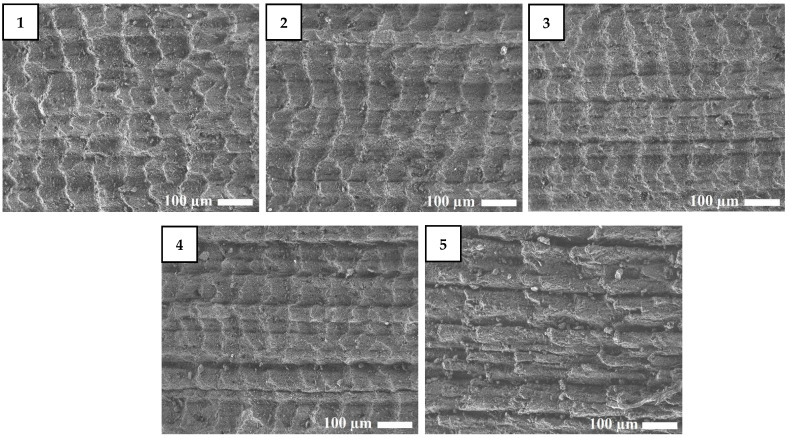
SEM images of rubber composites after abrasion (×150), **1**–**5**: corresponding rubber composite.

**Figure 3 molecules-26-00805-f003:**
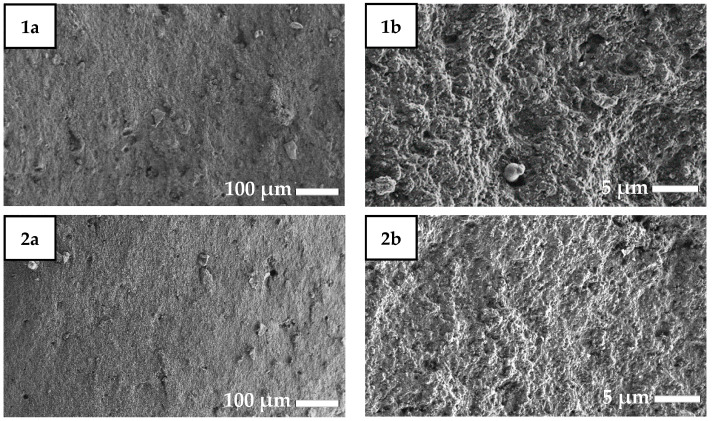
SEM images of ruptured rubber composites: **1**–**5**: corresponding rubber composite; (**a**): ×150 magnification; (**b**): ×3000 magnification.

**Table 1 molecules-26-00805-t001:** Vulcanization, physical, mechanical, and swelling properties of BNKS-18 rubber composites.

Properties	1	2	3	4	5
T_max_ (kPa)	1627	2244	2362	2444	1436
T_min_ (kPa)	310	302	305	302	263
ΔT (kPa)	1317	1942	2057	2142	1173
t_2_ (min)	0.68	0.32	0.28	0.26	0.76
t_90_ (min)	12.19	8.49	6.07	5.06	11.32
t_90_ − t_2_ (min)	11.51	8.17	5.79	4.80	10.56
CRI (min^−1^)	8.69	12.24	17.27	20.83	9.47
*f_P_* (MPa)	7.8 ± 0.5	17.5 ± 1.1	13.1 ± 0.8	11.4 ± 0.7	6.9 ± 0.4
*f_100_* (MPa)	2.1 ± 0.1	4.9 ± 0.3	5.6 ± 0.3	5.1 ± 0.3	1.7 ± 0.1
ε_p_ (%)	464 ± 29	306 ± 20	227 ± 14	203 ± 13	495 ± 30
*Q (%)*	9.69 ± 0.30	8.97 ± 0.28	8.42 ± 0.26	8.87 ± 0.27	9.51 ± 0.30
*Shore A*	68 ± 2	78 ± 2	79 ± 2	80 ± 3	68 ± 1

T_max_, T_min_: maximum and minimum torque, respectively; t_2_: scorch time; t_90_: optimum vulcanization time; CRI: cure rate index (100/(t_90_ − t_2_)), *f*_p_: tensile strength; *f_100_*: tensile stress at 100% elongation; ε_p_: elongation at break; Q: degree of swelling in Hydraunycoil FH51 oil at 70 °C for 72 h.

**Table 2 molecules-26-00805-t002:** Chemical composition of BNKS-18 rubber composites (phr).

Compounds	1	2	3	4	5
BNKS-18 base rubber	108	108	108	108	108
Carbon black N550	50	45	40	35	-
Activated carbon black N550	-	-	-	-	50
CPB-modified-carbon black N550 (CPB-CB)	0	5	10	15	0

## Data Availability

The data presented in this study are available on request from the corresponding author.

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
