# Peer review of "Enhanced Physical and Mechanical Properties of Nitrile-Butadiene Rubber Composites with *N*-Cetylpyridinium Bromide-Carbon Black"

_molecules, 2021, doi:10.3390/molecules26040805_

Round 1
Reviewer 1 Report
The manuscript has been improved it can be accepted.Reviewer 2 Report
The article "Enhanced Physical and Mechanical Properties of Nitrile-butadiene Rubber Composites with N-cetylpyridinium bromide-Carbon Black" is a revised version of already reviwed article by me. The Authors have supplemented required data and took into account my extensive remarks. It can be published in the present form.
Reviewer 3 Report
Thanks for your detailed response. Language and data presentation are fine now.
The only issue I have now is the statement on line 143:
"Fig. 1 (a) shows the slightly decreasing volumetric wear of the rubber composites with added 143 CPB–CB of up to 10 phr (composites 2–3), followed by an increase at 15 phr (composite 4)."
Given the errorbars the results are all the same and there is no trend to be seen.
This manuscript is a resubmission of an earlier submission. The following is a list of the peer review reports and author responses from that submission.
Round 1
Reviewer 1 Report
The manuscript presents interesting results regarding the experimental evaluation of using additives when synthesizing rubber composites. However, in my particular opinion, the manuscript should not be accepted for publication. Although it presents promising results, the authors should have used an experimental design technique, for example, Factorial design or Box-Behnken design, to adequately evaluate the additives' influence on the end-use properties of the composites. Besides, what is the experimental error of the results reported in Table 1? Duplicates or triplicates were carried out?
Additional experiments should be considered. A surface response could have been presented.
Reviewer 2 Report
The manuscript titled “Effects of cetyl-pyridinium bromide on 2 properties of rubber composites based on nitrile rubber” refers to the rubber industry's well-known use of multifunction compounds (MFA) as a substitution to the conventional activator/accelerators (for instance ZnO/stearic acid). In this particular case, the authors have mechanically activated carbon black with N-cetyl-pyridinium bromide (CPB). The MFA used is a surfactant compound with positively charged nitrogen at the end of its structure. Such a chemical compound structure has a significant effect on the scorch and cure rate of rubbers. With such a basis, one can think that the assumptions that the authors followed are correct and one can expect interesting, previously unpublished results.
The authors performed basic research on the characteristics of rubber mixtures and products, such as curing characteristics, physico-mechanical properties as well as structural characteristics. Such research, in the opinion of the reviewer, is sufficient to publish them in the journal. However, several aspects require major improvements and corrections.
- The language and grammar of the whole manuscript must be improved.
“Nitrile-butadiene rubbers with low content of acrylonitrile often used for the production of frost-resistant rubber composites.” – this is the first sentence of the article
“But CRI value of rubber composite with activated carbon black without CPB-CB is higher 116 than rubber 1 CRI only by 9%” – The sentence cannot start with “but”, please use something that fits scientific terminology like “however” or “although”.
Also, the whole article needs to be checked in terms of missing articles.
It is better to use the word “compounds” instead of “ingredients”
Please avoid repeating words in the same sentence
Please check punctuation for the whole manuscript
- The introduction must be supplemented with information such as: why a particular MFA is used (where the idea to use CPB comes from), any published research/books proving that cationic surfactants can actually work in the way the author expects them to.
- There is no explanation, how applied MFA works, what are expected reactions, and how does the MFA looks like in terms of chemical composition.
- “In previous work [12] investigation of carbon black mechanical activation in a planetary mill 55 and its influence on physical and mechanical properties of NBR rubber composites was conducted.” – The reference number 12 relates to “Okhlopkova, T.A.; Borisova, R.V.; Nikiforov, L.A.; Spiridonov, A.M., Okhlopkova, A.A.; Jeong, D.-Y.; Cho 224 J.-H. Supramolecular Structure and Mechanical Characteristics of Ultrahigh-Molecular-Weight 225 Polyethylene–Inorganic Nanoparticle Nanocomposites. Bull. Korean Chem. Soc. 2016, 37, pp. 439-444.” And it has nothing in common with NBR and carbon black.
- “As shown in Table 1, base rubber contained 100 parts per hundred parts of rubber (phr) of BNKS81 18 rubber, 2.5 phr of zinc oxide, 2,5 phr of sulfur, 1.5 phr of 2-mercapto-benzothiazole (MBT), 1,5 phr 82 of stearic acid as ordinary ingredients.” – In my opinion, it would be better to give the composition table in chapter 2, instead of referring to the table in the next chapter, leaving properties table in chapter 2. Moreover, table 1 does not show any “ordinary ingredients” mentioned in the text.
- All the figures given in the article are not affected by any error. The authors have not given any deviations, which is unfortunately not acceptable. The data must be supplemented with measurements deviations.
- How does the swelling degree was calculated? Moreover the authors investigated the influence of CBP on curing rate. While doing this it would be great to provide the sol fraction and swelling degree. It is a simple method but it helps to allay some of the doubts increasing the value of research work. It can be calculated by the Flory-Rehner equation with additional correction for samples containing fillers. The methodology can be found for example in: Marzocca, A.J. Evaluation of the polymer-solvent interaction parameter [chi] for the system cured styrene butadiene rubber and toluene. Polym. J. 2007, 43, 2682–2689 or Kim, D.Y.; Park, J.W.; Lee, D.Y. Correlation between the Crosslink Characteristics and Mechanical Properties of Natural Rubber Compound via Accelerators and Reinforcement. 2020.
- “Physical, mechanical and swelling properties of BNKS-18 rubber composites were determined 120 and summarized in Table 1.” – swelling should be counted as a physical property.
- “The degree of swelling for all BNKS-18 rubbers were almost constant, but the values tend to decrease with increase of CPB-CB content.” – that sentence is not true: 5% of CPB-CB gives 8.97, 10% gives 8.42 and for 15% the value rises up to 8.87. There is no decreasing tendency at all. With deviation, it is possible, that the values are similar.
- “CPB surfactant acts like dispersant and promotes the formation of new phase boundaries during joint activation with carbon black. “ - A very large contribution to the manuscript would be made by including schemes of the expected reactions or mechanisms, especially since the results obtained show a clear impact of modified carbon black. So far, the authors only assume how the reaction can proceed without giving up any sources confirming their assumptions.
- “During this process an intensive grinding of carbon black occurs with adsorption of surfactant on its surface. Before joint activation the surface of carbon black is hydrophobic. As a result of CPB adsorption a hydrophilization of the carbon black occurs. The surfactant-modified carbon black interacts better with more polar BNKS-18 rubber.” – Please provide references to confirm this.
- “Also, the presence of surfactant during activation causes lesser aggregation of carbon black particles after processing due to protected surface by surfactant.” – Was it confirmed by any microscopic examination, or is it another fact not supported by any reference?
- “This led to an increase of vulcanization rate, the tensile strength at 100% elongation and decrease of elongation at break. This can be explained by increased degree of crosslinking.” – look up point 6. It is not mandatory, but it is a good idea to confirm the degree of crosslinking with Flory-Rehner calculations.
- Figure 1 There is no deviation at all. Please provide it.
- How does the coefficient of friction was calculated? What was the method? Was it conducted on stainless steel? Dry or wet? Is there any explanation why the coefficient changes with the visible trend (decreasing with increasing content of modified carbon black), while volumetric wear changes by a maximum of 0.015 cm3?
- “We investigated effects of CPB as a modifier of carbon black by means of joint activation in a planetary mill.” – It is better to write an article in an impersonal form.
- “Addition of CPB-CB to BNKS-18 rubber composites improves the physical and mechanical, swelling, tribological properties. CPB causes a hydrophilization of carbon black which leads to its better interaction and distribution with rubber macromolecules. Also, the surfactant acts like vulcanizing agent and improves vulcanization characteristics of the rubber composites” – I don't understand why the authors did not prepare a sample that would contain CBP added with the curing system. It would allow verifying immediately the validity of carbon black modification by the mentioned MFA and it use in rubber compositions. Right now it is unknown if the mechanical activation was necessary, especially if the sample with unmodified CB has almost identical physico-mechanical properties as a sample with activated CB (tensile strength – 7.8 and 6.9 MPa, elongation at break 464 and 495, swelling 9.69 and 9.51 % and hardness 68 and 68 Sh A, respectively for sample 1 and 2).
- “Thus, we have developed a new class of modifiers of elastomeric materials, combining the properties of dispersants and vulcanizing agents. Small concentrations of the modifier, which are necessary to significantly improve the technological and properties of rubbers, should not significantly affect the cost of modified materials. “ - Have preliminary calculations been made of the costs of modifications together with the costs of additional treatment such as carbon black activation?
- The abstract starts with the sentence: “Nitrile-butadiene rubbers with low content of acrylonitrile often used for production of frost resistant rubber composites”. Also, the first paragraph of the introduction tells the readers about frost resistance. After that, there is no single word about how the proposed modification can influence this parameter. There is any conclusion saying for example that it can enhance the properties of products used in low temperatures. So what was the reason for raising the subject of resistance to low temperatures?
In general, the whole paper requires MAJOR corrections and every comment must be addressed and updated within the manuscript. The approach is interesting, and the results indicate that the applied MFA actually influences the vulcanization of rubber compositions. However, the manuscript cannot be published in the present form.
Reviewer 3 Report
EFFECTS OF CETYL-PYRIDINIUM BROMIDE ON PROPERTIES OF RUBBER COMPOSITES
BASED ON NITRILE RUBBER
General impression: The paper presents mechanical and microscopic characterization of low-ACN NBR compounds, mixed with an increasing amount of surface-modified standard carbon black. The idea of surface-modifiying carbon black to compatibilize it with the quite polar NBR certainly is of industrial interest. However, the paper is rather poorly made: Language needs more precision, experimental results give no hint of their reproducibility and the selection of experiments appears random.
In more detail:
• Please revise language to be more precise and grammarly correct. Examples:
o Line 39: “and turn loose polymer molecules into more rigid and reinforced matrix” ?
o Line 44: “Park and Kim used a method of chemical treatment …” this can be everything, be more precise.
o Please restate lines 139-144. What’s “total vulcanization”?
• Line 90: Why do you add the CPB-CB on the roller mill? Why is there more scorch using the CPB-CB? Try to explain. In this context: One of your main conclusion is that the CPB helps to achieve good dispersion. Is it possible to measure the latter?
• Perhaps I missed it, but what exactly is “Activated carbon black”? Just the carbon black put into the planetary mill without the surfactant?
• Why don’t you show the vulcanization curve? This gives way more information than all the values you supply. Moreover: As there is so much space in the table, please help the reader by writing the actual meaning of the many variables in front of the variable. (e.g. “tensile strength fP [MPa]”).
• From my experience tensile strength of compound 1 should be much higher. Perhaps there is poor dispersion due to insufficient mixing? This is just a guess, I may be wrong here.
• Show tensile curves (Strain/stress)
• Please avoid definitions of abbreviations in labels (“CRI”, line 110).
• Figure 1: What’s the reproducibility of all the results (errorbars)?
• What’s the substrate you are using to determine the friction coefficient? A mu of 2 suggests steel?
• Please supply a scale for the SEM images given in Fig. 3
Overall, I cannot recommend publication in the present form. It may be suitable for publication after the issues above were addressed.
Reviewer 4 Report
the authors should make more clear the comparison of the sample with optimum composition (sample 2) with the rubber composite which contains only initial carbon black (sample 1).